# Architectures and Applications of BODIPY-Based Conjugated Polymers

**DOI:** 10.3390/polym13010075

**Published:** 2020-12-27

**Authors:** Yiqi Fan, Jinjin Zhang, Zhouyi Hong, Huayu Qiu, Yang Li, Shouchun Yin

**Affiliations:** 1College of Material, Chemistry and Chemical Engineering, Hangzhou Normal University, Hangzhou 311121, China; fanyiqi@stu.hznu.edu.cn (Y.F.); zhangjinjin@stu.hznu.edu.cn (J.Z.); 2018211705173@stu.hznu.edu.cn (Z.H.); hyqiu@hznu.edu.cn (H.Q.); 2Key Laboratory of Organosilicon Chemistry and Materials Technology of Ministry of Education, Hangzhou Normal University, Hangzhou 311121, China

**Keywords:** BODIPY, conjugated polymers, architecture, structure–property relationship, application

## Abstract

Conjugated polymers generally contain conjugated backbone structures with benzene, heterocycle, double bond, or triple bond, so that they have properties similar to semiconductors and even conductors. Their energy band gap is very small and can be adjusted via chemical doping, allowing for excellent photoelectric properties. To obtain prominent conjugated materials, numerous well-designed polymer backbones have been reported, such as polyphenylenevinylene, polyphenylene acetylene, polycarbazole, and polyfluorene. 4,4′-Difluoro-4-bora-3a,4a-diaza-s-indacene (BODIPY)-based conjugated polymers have also been prepared owing to its conjugated structure and intriguing optical properties, including high absorption coefficients, excellent thermal/photochemical stability, and high quantum yield. Most importantly, the properties of BODIPYs can be easily tuned by chemical modification on the dipyrromethene core, which endows the conjugated polymers with multiple functionalities. In this paper, BODIPY-based conjugated polymers are reviewed, focusing on their structures and applications. The forms of BODIPY-based conjugated polymers include linear, coiled, and porous structures, and their structure–property relationship is explored. Also, typical applications in optoelectronic materials, sensors, gas/energy storage, biotherapy, and bioimaging are presented and discussed in detail. Finally, the review provides an insight into the challenges in the development of BODIPY-based conjugated polymers.

## 1. Introduction

In the 1970s, Shirakawa et al. accidentally synthesized polyacetylene and then developed its conductive applications under close cooperation with MacDiarmid and Heeger [1,2,3,4]. Since conductive polymers emerged, conjugated polymers, namely, polymers with conjugated bond structures, have attracted increasing interest owing to their extremely important roles in optoelectronic devices [5,6,7] and biological diagnosis and treatment [8,9]. In general, the energy band gap of conjugated polymers is very small and can be adjusted via chemical doping to obtain excellent optoelectronic properties. Furthermore, conjugated polymers offer a unique combination of properties not available from other materials, such as mechanical flexibility, facile solution processing, scalable architecture, and low cost. To develop conjugated materials with better performance, a large number of functional units have been successfully introduced to polymer systems, including phenylacetylene [10], thiophene [11], carbazole [12], fluorene [13], and 4,4′-difluoro-4-bora-3a,4a-diaza-s-indacene (BODIPY) [14]. Among them, BODIPY (Figure 1) is famous for its versatility as fluorophores and is widely applied in fluorescent sensors, organic electronics, biotherapy, and imaging [14,15,16,17,18], which provides a possibility for the multifunctional development of conjugated polymers.

In 1968, BODIPY was reported for the first time by Treibs and Kreuzer [19], and later, Hee and Richard discovered that BODIPY could be used as a fluorescent probe with high affinity for D_1_ and D_2_ dopaminergic receptors [20]. Since then, BODIPYs have gained increasing attention in most fields. In most cases, BODIPYs have excellent optical properties, including large molar extinction, narrow absorption, and high quantum yields. BODIPYs are also stable to thermal/photochemical stimulus and possess good solubility to common solutions, which is conducive to the processing and use of optoelectronic devices. Most importantly, the optical properties and electronic structure of BODIPYs can be easily adjusted via systematic structural modifications [21,22]. For instance, modifying the BODIPY core with donor units at 2,3,5,6-positions would cause a bathochromic shift in the absorption and fluorescence spectra. All the advantages of BODIPY make it a promising candidate for the preparation of conjugated polymers with innovative molecular structures and unique electronic properties [22].

Due to the diverse strategies for the functionalization of the BODIPY, several synthesis methods have been presented to establish BODIPY-based conjugated polymers through a wide variety of BODIPY intermediates (Figure 1) with reactive groups (halogen, alkynyl). Taking halogenated BODIPYs as an example, halogen atoms can be easily introduced into BODIPY core in α-, β-, or meso-positions via electrophilic substitution reactions or using halogenated dipyrromethene precursors [14]. Subsequently, halogenated BODIPYs can homopolymerize by nickel-catalyzed Yamamoto cross-coupling reaction or undergo alternating copolymerization by palladium-catalyzed cross-coupling reactions, including Suzuki, Heck, Sonogashira, and Stille cross-coupling reactions. Without halogen atom, BODIPY-based conjugated polymers could be obtained by Oxidation and Friedel-Crafts cross-coupling reactions and introducing alkynyl in α- and β-positions of BODIPY cores to support Sonogashira cross-coupling reactions are also feasible methods. Additionally, BODIPYs combined with electroactive 3,4-ethylenedioxythiophene (EDOT) or bithiophene can be prepared via electrochemical polymerization, as reported by Algi, Cihaner, and Skabara [23,24,25].

Current studies have found that by incorporating BODIPYs into the constructions of conjugated polymers, the corresponding polymers inherit the excellent optical properties of BODIPY due to the increase in conjugated length [14,26,27,28]. For example, to study the correlation between the number of BODIPY molecules and the polymer properties, Allen and coworkers [26] synthesized a monomer, dimer, trimer, and polymer of BODIPYs with mesityl groups in the meso-position, and the photophysical, electrochemical, and electrogenerated chemiluminescence of the obtained materials were investigated. The absorption and fluorescence of the materials showed a gradually red shift from monomer to dimer to trimer to polymer, accompanied by increasing molar absorption coefficients and decreasing quantum yields [26].

In this review, we focus on the ample structures of BODIPY-based conjugated polymers (including linear and porous structures) based on different reaction sites and functionality degrees of monomers. Special attention is devoted to the applications of conjugated polymers containing BODIPYs in optoelectronic materials, biotherapy, bioimaging, sensors, and gas/energy storage, wherein the structure–property relationship exerts tremendous influence.

## 2. Structures of BODIPY-Based Conjugated Polymers

According to the functionality degree, structures of BODIPY-based conjugated polymers can be classified into linear/coiled, porous, dendritic, and crosslinked. Thus, in the following subsections, the synthesis, property, and structure–property relationship of BODIPY-based conjugated polymers are discussed from a structural perspective. Some structures and synthetic methods of BODIPY-based conjugated polymers are summarized in Table 1.

### 2.1. Linear/Coiled Conjugated Polymers

Generally, the monomers forming a linear/coiled polymer should have two functional groups, such as halogen atoms and alkynyl. In the constructed conjugation of linear/coiled polymers, BODIPYs can be incorporated into the polymers as backbone units, pendant side chains, or end groups.

#### 2.1.1. BODIPY as Backbone Units

When BODIPYs act as the backbone units, the structure and properties of conjugated polymers are prone to site-selective synthesis; that is, polymerization through either α- or β-positions. The β-connected copolymers possess linear structures, whereas the α-connected ones possess coiled structures [29]. Zade and coworkers [30] presented three alternate copolymers, **1–3** (Figure 2), containing BODIPYs and acetylene with different connectivities (α-α-connected, α-β-connected, β-β-connected) and examined the effect of site-selective copolymerization. The decomposition temperatures of polymers **1**, **2**, and **3** were 266, 312, and 200 °C respectively, because of the difference in the compactness of the polymer melt. In other words, the coiled structures contributed to increasing the decomposition temperature. In addition, the solid-state optical band gaps of **1**, **2**, and **3** were 1.28, 1.42, and 1.67 eV respectively, which demonstrated that the π-conjugation was more efficient in the case of polymerization through the α-position of BODIPYs than in the polymerization through the β-position. The site-selective synthesis mainly altered the highest occupied molecular orbital (HOMO) energy levels of polymers rather than the lowest unoccupied molecular orbital (LUMO) energy levels. Based on the theoretical calculation results, polymers **1–3** were subjected to charge-carrier mobility measurements in field-effect transistors devices, in which **1** and **3** acted as n-type semiconductors and **2** as a p-type semiconductor. However, their transistor property was poor. Nevertheless, this work provided an efficient approach for tuning the band gaps of BODIPY-based conjugated polymers via site-selective polymerization. Likewise, Zade and coworkers [31] synthesized polymers **4–6** (Figure 2) with BODIPYs and 5,5-bis(hexyloxymethyl)-5,6-dihydro-4H-cyclopenta[c]-thiophene (CPT) as comonomers. The CPT was connected through the α-position or β-position of the BODIPY molecule, and **6** had acetylene as the spacers. Polymer **4** exhibited the highest *p*-channel mobility in polymer field-effect transistors, attributed to the efficient conjugation of the α-connectivity of BODIPYs as well as their planar structure without methyl at β-positions. Due to the existence of the spacers, **6** with planarity of the backbone showed the second-highest host mobilities. Except for charge mobility, the BODIPYs with connectivity at α-position may produce better photovoltaic properties [29] and more intriguing nonlinear optical properties [32].

Apart from site-selective copolymerization, another similar synthesis method is site-selective modification with polymerizable groups before polymerization. Three polymers, **7–9** (Figure 3), have been constructed through the electropolymerization of BODIPYs functionalized by EDOT at 3,5-positions or 2,6-positions. Polymer **8** modified through 3,5-positions showed a bathochromic shift of the absorption wavelength and a decrease in band gaps than those of polymers **7** and **9** modified through 2,6-positions, which further proved that the connected positions resulted in significant differences in the conjugation degree. Polymer **8** displayed the lowest band gap of 0.8 eV, with reversible reduction and oxidation processes [23,25,33].

The properties of BODIPY-based conjugated polymers also depend on the elaborate design of BODIPY monomers with functional modification. For instance, the incorporation of oligo(ethylene glycol)methyl ether residues as polymer side chains into BODIPYs at the 3,5- or meso-positions led to a significant increase in water solubility [34]. In another work, Liu and coworkers [35] presented a series of near-infrared (NIR) or deep-red emissive polymeric dyes **10**–**14** (Figure 4) containing extended π-conjugated BODIPY cores. Polymers **10** and **11** were synthesized via the Sonogashira cross-coupling reaction of 2,6-diethynel BODIPYs with 2,6-diiodo-functionalized BODIPYs bearing aldehyde derivative-modified mono-styryl or di-styryl groups at the 3,5 positions. For better comparison, **12**–**14** were prepared by the same reaction of 2,6-diiodo-functionalized BODIPYs bearing mono-styryl group, di-styryl group, and dodecyl with 2,5-diethynyl-3-decylthiothene, respectively. The fluorescence quantum yields of **12–14** were significantly lower than those of their analog polymers **10** and **11**, due to the heavy atom effect of sulfur. In solutions, the absorption and emission maxima of **11** were centered at 738 and 760 nm respectively, with redshifts of 41 and 45 nm compared with the spectrum for **10**. Similarly, by increasing the number of conjugated connections at the 3,5-positions of BODIPY cores, the absorption and emission maxima of **14**, **13**, and **12** successively redshifted, which allowed the tuning of BODIPY-based polymers to the NIR region. As for thin films, the absorption and emission maxima of **10–12** displayed further redshift, attributed to the intermolecular electronic interaction and/or increasing planarity in the solid state. Remarkably, the extension of conjugated BODIPYs at 3,5-position resulted in deep-red or NIR emission [36,37], and such NIR-emissive BODIPY-containing polymers lay a foundation for the development of NIR-imaging biomedical applications.

BODIPY-fused aromatic rings such as phenyl [38], furan [39], and thiophene [40] are promising candidates for performance development of BODIPY-containing polymers. Chujo and coworkers [40] reported two conjugated homopolymers **15** and **16** (Figure 5) consisting of thiophene-fused BODIPYs and synthesized via oxidative coupling. With the suppression of torsions, low-laying HOMO, small band gap, and rigid framework, the thiophene-fused BODIPYs endowed the polymers with low band gap and high stability in atmosphere and a broad absorption spectrum. Polymer **16** showed wider absorption and lower band gaps than the methyl-substituted analog **15** due to the smaller steric hindrance. Such ring-fused BODIPYs can open new approaches for the construction of low-band-gap conjugated polymers with broad absorption and good stability.

The most common and patterned approach to developing novel polymer materials containing BODIPYs is selecting the appropriate comonomers to facilitate intra-molecular charge transfer excitation [41,42,43,44,45,46,47]. To precisely predict the performance of alternating copolymers containing BODIPYs, Thayumanavan and coworkers [48] chose five aromatic groups with different electron-donating capacity (including fluorene (**17**), carbazole (**18**), bithiophene (**19**), cyclopentadithiophene (**20**), and dithienopyrrole (**21**)) to copolymerize with BODIPYs and investigated the influence of comonomers on the properties of the ensuing polymers **17**–**21** (Figure 6), especially the frontier molecular orbital energy levels. The HOMO energy levels of polymers, assessed by density functional theory (DFT) calculations or cyclic voltammetry, showed the same tendency as their gas-phase ionization potential (IP) values. In contrast, the LUMO energy levels were almost invariant, which indicates that the influence of BODIPY on HOMO energy levels was dominant. The present study builds up the key link of polymer design with theoretical predictions, which facilitates the development of BODIPY-based conjugated polymers.

Samuel and coworkers [49] reported two low-band-gap donor-acceptor (D-A) conjugated polymers, **22** and **23**, containing BODIPY as electron acceptors and bis(3,4-ethylenedioxythiophene) (bis-EDOT) and bis(3,4-ethylenedithiathiophene) (bis-EDTT) as electron donors, respectively (Figure 7). In the ultraviolet-visible (UV-vis) absorption spectra measured in solution, **22** displayed broad absorption from about 300 to 1000 nm with maxima at 818 nm, while **23** showed a narrower absorption from around 400 to 900 nm with maxima at 648 nm. This suggested that bis-EDOT possessed stronger donor ability than bis-EDTT. Calculated from the absorption spectra, the optical band gaps of **22** and **23** were 1.18 and 1.35 eV, respectively. Furthermore, the time-of-flight measurements proved that **22** allowed for ambipolar charge transport. Two types of devices with the obtained polymers as electron donors and [6,6]-phenyl-C_71_-butyric acid methyl ester (PCBM) as the electron acceptor were separately fabricated. When the donor-to-acceptor ratio was 1:4, the device made of **22** exhibited the best performance: a circuit current density (*J*_sc_) of 7.78 mA cm^−2^, open-circuit voltage (*V*_oc_) of 0.31 V, fill factor (FF) of 39%, and power conversion efficiency (PCE) of 0.95% were observed. Clearly, the BODIPY-based conjugated polymer is a promising material for improving the performance of polymer-fullerene devices by rationally decreasing band gaps. 

#### 2.1.2. BODIPY as Pendants Side Chains

Generally, conjugated polymers with BODIPYs as pendants side chain were synthesized through meso-positions connection of BODIPYs containing reactive groups, such as alkynyl [50] electrochemical polymer groups [51,52] or other more complex units [53]. Chujo and coworkers [54] tethered BODIPY dyes to the cardo structures of polyfluorenes in order to achieve advanced light-harvesting antenna (LHA) systems (Figure 8). Polymer **24** was synthesized via the polymerization of mono-BODIPY substituted dibromo cardo fluorene derivates through Yamamoto cross-coupling reaction, while polymers **25** and **26** were prepared via the combination of 9,9-didodecylfluorene diboronic acid with dibromofluorene containing single or dual BODIPYs at its cardo structures through Suzuki-Miyaura cross-coupling reaction. By comparing the absorption and photoluminescence spectra of polymers and simple mixtures of free BODIPY units and polyfluorene, it was found that the intrinsic optical properties of BODIPYs and polyfluorene main chains were retained. Moreover, **25** and **26** exhibited significant fluorescence emissions in both solution and film states, due to the efficient suppression of the concentration quenching of BODIPYs. The LHA efficiency of **25** was nine times that of the free BODIPY unit, and the energy transfer efficiencies of **24**–**26** were up to 99% because of the rigid cardo structures of polyfluorene. Based on their excellent properties and good processability, these effective LHA systems with BODIPY-attached cardo polyfluorenes are promising for practical applications.

Additionally, BODIPYs can be incorporated into conjugated polymers as pendant side chains through meso- or α-positions. Yin and coworkers [55] synthesized three polyacetylenes **27**–**29** bearing BODIPYs as pendant side chains through meso- or α-positions (Figure 9) and investigated the effect of connective methods on the properties of the obtained polymers. Polymers **28** and **29** displayed poorer thermal stability than **27** because the coplanarity of backbone and BODIPY units in polymers **28** and **29** reduced the thermal resistance of polymers. Moreover, **28** and **29** had a broader absorption range, along with a little redshift, than corresponding monomers, while the absorption spectrum of **27** was similar to that of the monomer owing to a vertical arrangement of BODIPY units with main backbone. All polymers exhibited better nonlinear optical properties than polyacetylene. Furthermore, the third-order nonlinear optical coefficients of **28** (1.27 × 10^−10^ esu) (esu is abbreviation for electrostatic unit) and **29** (1.37 × 10^−10^ esu) were ~15 times larger than those of **27** (8.5 × 10^−12^ esu). Accordingly, the connective methods of BODIPYs had a significant influence on the properties of the ensuing polyacetylenes, and the desired photoelectric properties could be achieved by the attachment of polymerizable groups to BODIPYs at the optimal positions.

#### 2.1.3. BODIPY as End Groups

To gather the schematic information on the relationship of energy transfer and rigid chain, Yin and coworkers designed a novel poly(*p*-phenylene ethynylene) (**30**–**19**) [56] and a group of oligo(*p*-phenylene ethynylene)s (**30-1, 30-3, 30-5, 30-7**) [57] with different conjugated lengths (Figure 10)—they were capped with BODIPYs at their ends. According to the absorption, excitation, and emission spectra, the synthesized polymer or oligomers inherited the spectroscopic properties of both BODIPYs and phenylacetylene chain. Both the absorption and emission maxima assigned to the phenylacetylene main chains redshifted as the number of phenylacetylene units was increased, however, those assigned to the BODIPY units had almost no change and the relative intensity of the absorption was decreased. The energy transfer efficiency reduced as the conjugated length increased because of the decrease in the energy transfer rate and mass of conformational subunits. Thus, this work showed that the energy transfer efficiency of dye-capped polymers could be tuned by changing the conjugated chain length.

### 2.2. Porous Conjugated Polymers

Porous organic polymers (POPs) with conjugated structures have attracted pronounced attention owing to the high flexibility in their skeleton design and porosity; consequently, their applications have been extended to gas absorption and separation, energy storage, catalysis, and optoelectronic materials [58,59,60,61,62,63,64]. Especially, with facile modification, diversity of reaction sites, and intriguing optical properties, BODIPYs have been widely used as skeleton units of POPs. Algi and coworkers [24] designed two BODIPY cores, bithiophene and EDOT, with three electroactive function groups, and the ensuing polymers **31** and **32** (Figure 11) obtained via electrochemical polymerization possessed cellular structures and multi-electrochromic properties.

Kuang and coworkers presented a series of BODIPY-based POPs **33**–**37** (Figure 12) synthesized by Friedel-Crafts cross-coupling reactions to explore the influences of monomer structures on the porous properties and singlet-oxygen generation properties [65,66]. In comparison with analogs **33**–**35**, with a long space between two BODIPY units, showed higher surface area but worse photocatalytic activity of singlet-oxygen generation. Although **34** and **36** with or without methyl substituents in the BODIPY cores possessed good pore characteristics, **34** presented the coexistence of micropores and mesopores, and **36** only presented mesoporous architecture. Additionally, the singlet-oxygen generation ability of **34** was better than that of **36**. Compared with **36**, **37** showed poor pore properties and singlet-oxygen generation capacities, implying that the monomeric isomerization was a crucial parameter for the design of BODIPY-based POPs. This work about the relationship between monomeric structures and characteristics can lay a foundation for designing more intriguing BODIPY-based POPs.

### 2.3. Other Structures

Apart from the above-mentioned categories, some BODIPY-based conjugated polymers also contain branched polymers and crosslinked polymers, whose syntheses and applications are introduced in Section 3.

## 3. Functional Applications of BODIPY-Based Conjugated Polymers

BODIPY-based conjugated polymers are fascinating in that they present a perfect fusion of π-conjugated backbone and unique BODIPY dyes. Furthermore, the precise control of structures can be achieved at the molecular level to realize the desired performance. By virtue of various morphologies and excellent photoelectric properties, conjugated polymers containing BODIPYs have emerged as versatile materials for applications ranging from optoelectronic materials, biotherapy and imaging, sensor, gas adsorption, and energy storage. The structures, applications, and synthetic methods of BODIPY-based conjugated polymers are summarized in Table 1 and Table 2.

### 3.1. Optoelectronic Materials

Conjugated polymers have always been one of the main optoelectronic materials owing to their fine-tuning optoelectronic properties based on the ample electron-rich and electron-deficient aromatics. BODIPYs, classical acceptors, possess excellent photophysical and electrochemical properties, endowing BODIPY-based conjugated polymers with extensive application in optoelectronics, such as solar cells, organic thin-film transistors, and memory devices.

To endow solar cells with excellent performance, the molecular design of BODIPY-based conjugated polymers are the following three strategies: (1) the introduction of electron-rich comonomers during polymerization [67,68,69,70,71,72], (2) fusing aromatic rings on the BODIPY units [73], and (3) the extension of the π-conjugation of side chain [73,74]. Donor-acceptor polymers tend to be utilized as donor materials, and fullerene derivatives such as phenyl-C_61_-butyric acid methyl ester (PC_71_BM) as acceptor materials, allowing for the construction of solution-processed bulk heterojunction (BHJ). In 2015, Chochos and coworkers [75] reported a novel ultra-low-band-gap and NIR conjugated polymer **38** (Figure 13), which was synthesized by dibromo-BODIPYs and (*E*)-1,2-bis(3-dodecyl-5-(trimethylstannyl)thiophen-2-yl)ethane via Stille cross-coupling reaction. The obtained polymer displayed a panchromatic absorption in the 300–1100 nm range with an optical band gap of 1.15 eV, promoting the construction of NIR BHJ solar cells based on **38** and PC_71_BM. When the weight ratio of **38** to PC_71_BM was 1:3, the solar cell showed the best PCE of 1.1% with *J*_sc_ of 3.39 mA cm^−2^, V_oc_ of 0.59 V, and FF of 0.56, while the highest charge carrier mobility was (1.05 ± 0.2) × 10^−5^ cm^2^V^−1^s^−1^ for the 1:4 ratio. In 2008, Sharma and coworkers [74] also presented BJH solar cells based on PC_71_BM, wherein low-band-gap conjugated polymers **39** and **40** (Figure 13) served as electron donors. Polymer **39** with BODIPYs, thiophene unites, and ethynyl linkers displayed an absorption spectrum ranging from 500 to 800 nm and an optical band gap of 1.74 eV. Polymer **40** with different extended BODIPY unit was achieved by introducing Zinc(II) porphyrins into BODIPY units of **39** via Knoevenagel condensation. Owing to the extension of conjugation, **40** exhibited broadened absorption, ranging from 400 to 800 nm, and decreased optical band gap (1.59 eV). Bulk heterojunction solar cells with **39**/PC_71_BM and **40**/PC_71_BM ratio of 1:2 showed PCE of 3.03% and 3.86%, respectively. The above two works demonstrated that BODIPYs are promising building blocks for devising novel polymeric donors for solar cells conjugated with PC_71_BM.

There are also many organic solar cells consisting of BODIPY-based conjugated polymers as donors and non-fullerenes as receptors. Sharma and coworkers [76] developed **41** (Figure 14), which is an analogous BODIPY-based conjugated polymer of **39**, with monosubstituted thiophene units via Sonogashira cross-coupling reaction. The photovoltaic properties of BHJ solar cell based on **41** as the donor and non-fullerene small molecule (NFSMA) as the acceptor were compared with that of the BHJ solar cell with PC_71_BM as the acceptor. Polymer **41** and NFSMA exhibited complementary absorption favorable to light harvesting and well-matched frontier energy level favorable to efficient charge transfer and minimal energy loss. In addition, Li and coworkers [77] fabricated all-polymer solar cells, in which BODIPY-based conjugated polymers without (**42**) or with (**43**) methyl substituent on the BODIPY cores served as donors, and polymer N2200 consisting of bithiophene and hexahydropyrene derivative served as acceptors (Figure 14). Notably, the complementary interaction between acceptor and donor led to the absorption spectrum extension from 300 to 900 nm in solar cells. After experimental analysis, it was found that **43** with two methyl substituents displayed superior photovoltaic properties compared to polymer **42** without methyl substituent. On account of neat structures, suitable energy levels, and high hole mobility, the PCE of the **43**-based solar cell was 5.8%, higher than that of the **42**-based solar cell (0.32%). Considering the superiority of **43**, Li and coworkers [78] further studied the performances of solar cells composed of electron donor **43** and halogenated fused-ring electron acceptor ITIC-2Cl or BTP-2Cl (Figure 14). Likewise, a wide absorption spectrum from 300 to 1100 nm was obtained, and the external quantum efficiencies were above 0.50. Interestingly, a dramatically high *J*_sc_ (21.44 mA cm^−2^) was obtained from the BTP-2Cl-based solar cell, and the PCEs increased to 7.56% and 9.86% for ITIC-2Cl- and BTP-2Cl-based solar cells, respectively. Overall, the rational molecular design of BODIPY-based conjugated polymers offers advantages in terms of PCE enhancement, and it is also a good idea to match the polymer acceptor or small-molecule acceptor, replacing fullerene receptors.

Perovskite solar cells (PSCs) are considered typical representatives of the third generation of solar cells owing to their high efficiency, low cost, and easy preparation [79,80]. However, their stability could be one of the obstacles limiting their development. Consequently, Hong and coworkers [81] developed a series of dopant-free hole-transport materials (HTMs) based on BODIPY-containing conjugated polymers **44**–**49** (Figure 15) to improve the PSC stability. The selective regulation of HOMO/LUMO value was achieved by introducing groups at the positions of BODIPY cores and benzo[1,2-*b*:4,5-*b’*]bithiophene. Owing to solubilizing groups and nonplanar structure, hydrophobic HTM polymers could entirely cover the perovskite layer, allowing it to isolate air and moisture. Because of different HOMO/LUMO energy levels and electrical properties, **44**–**49** exhibited different performances in solar cells, among which **44** displayed the highest PCE of 16.02%, with *J*_sc_ of 19.22 mA cm^−2^, V_oc_ of 1.06 V, FF of 0.78, and hole mobility of 2.96 × 10^−5^ cm^−2^ V^−1^ s^−1^. The device stability test showed that **44**-based PSC retained 80% of the original PCE under ambient condition after 10 days.

Apart from solar cells, organic thin-film transistors (OTFTs) are another application of BODIPY-based conjugated polymers in the area of optoelectronic materials [82,83,84,85,86]. Facchetti and coworkers [82] designed a group of D-A copolymers **50**–**53** consisting of alternating tetramethyl-substituted or dimethyl-substituted BODIPYs and thiophene or 3,3′-dialkoxy-2,2′ bithiophene bridged by Stille cross-coupling reaction (Figure 16). The polymers as thin-films displayed panchromatic absorption spectra up to the NIR region and 60–90 nm redshift compared with the corresponding solution spectra. According to experimental estimation, HOMO energies were determined as −5.04 to −5.59 eV for **50**–**53**. In addition, polymers based on dimethyl-substituted BODIPYs conjugated with thiophene units possessed better planarity and more efficient π-stacking, leading to more potent intra- or inter-chain charge transport. Hence, it was not surprising that in the OTFTs, **52** exhibited *p*-channel activities with the highest hole mobility of 0.17 cm^−2^ V^−1^ s^−1^ and on/off ratios of 10^5^–10^6^, and favorable stability. Moreover, recently, Sharapov and coworkers [83] presented a series of BODIPY-based conjugated polymers **54**–**56** containing thiophene-fused BODIPY units and donor groups such as fluorene, BDT, and diketopyrrolopyrrole (DPP) to develop OTFTs (Figure 16). The absorption range of the polymers widened in turn with the maximum absorption redshifted. Particularly, **56** possessed a broad and strong absorption in the range of 500–1600 nm. The cyclic voltammetry results revealed low LUMO energy levels (<−4.0 eV) of polymers, resulting in favorable electron injection and transport. The electron mobilities of **54**, **55**, and **56** were 4.0 × 10^−5^, 7.8 × 10^−4^, and 5.4 × 10^−4^ cm^−2^ V^−1^ s^−1^, respectively. However, due to inadequate HOMO energy level, hole mobility was not observed for **54**, while **55** and **56** with lower HOMO energy levels possessed hole mobilities of 1.0 × 10^−3^ and 7.2 × 10^−4^ cm^−2^ V^−1^ s^−1^, respectively. Furthermore, polymer **56** displayed ambipolar charge mobility of 10^−3^ cm^−2^ V^−1^ s^−1^ in OTFE devices, which is a good parameter for BODIPY-based conjugated polymers. BODIPY-based conjugated polymers also presented potential to serve as n-type semiconducting polymers by the judicious choice of comonomers; for example, the alternating copolymer **57** has been reported by Thayumanavan and coworkers [84] (Figure 16). Thus, BODIPY-based conjugated polymers could play versatile roles in the design of OTFTs.

Chen and coworkers [87] combined BODIPY units and 1,4-diethynylbenzene with reduced graphene oxide (RGO) via one-pot in situ polymerization to produce the two-dimensional BODIPY-based conjugated polymer **58** (Figure 17). RGO served as an electron donor and BODIPY-contained polymeric chain served as an electron acceptor. The product exhibited good storage performance of rewritable memory that can be electrically erased and reprogrammed, without disturbance by mechanical stress. This may be caused by the efficient conjugated channel and the favorable charge transfer from RGO to **58**. This work provides an example of a stable high-performance memory device, which not only extends the applications of BODIPY-based conjugated polymers but also presents a hint for improving the stability of memory devices.

### 3.2. Bioimaging and Biotherapy

A significant amount of effort has been channeled toward the development of specific imaging and effective therapies, where BODIPY-based conjugated polymers have ample scope due to their high expansibility of optical properties [64,88,89,90,91,92,93,94,95]. Fluorescence imaging can help to provide visualized results with simple operation and low cost. Compared with visible light, NIR has been a research hotspot in the field of fluorescence imaging owing to the high penetration depth, small biological spontaneous fluorescence interference, and small light-damage to organisms. Liu and coworkers [92] reported a NIR-emissive BODIPY-containing polymer **60** bearing cancer-homing cyclic arginine-glycine-aspartic acid (RGD) peptide residues. The polymer was synthesized via the post-polymerization functionalization of **59** through click reaction between sulfydryl and halogen (Figure 18). Driven by the existence of tetra(ethylene glycol), **59** and **60** exhibited good water solubility and good photostability, conducive to applications in organisms. Moreover, **59** and **60** showed similar absorption and emission spectra, with the absorption maximum at 687 (**59**) and 689 nm (**60**) and emission maximum at 711 (**59**) and 712 nm (**60**). Polymer **60** displayed enhanced NIR signals at breast cancer cells in the perinuclear region, which indicated specific recognition with breast cancer cells. Thus, this research not only presents a NIR-emissive BODIPY-based polymeric dye for specific cell imaging but also provides a basis for designing specific imaging agents through binding with targeted groups.

Water solubility is another important favorable factor for the application of polymeric dye in organisms. In addition to the aforementioned introduction of hydrophilic chains, ionization is also a feasible strategy. Cao and coworkers [95] designed a water-soluble conjugated polyelectrolyte **61** with BODIPYs and quaternary ammonium salt-modified fluorenes as monomers (Figure 19). Polymer **61** possessed distinct “turn-off” fluorescence characteristics to DNA due to the electrostatic interaction of positively charged polyelectrolyte and negatively charged DNA of **61**-DNA complexes, which allowed it to be a detector agent for DNA. Furthermore, red emission was observed when Hela cells were incubated with 0.1 mg/mL **61**, which suggests the potential of **61** for bioimaging applications.

Conjugated polymer dots (Pdots) are capable candidates for biological applications, with various advantages such as high brightness, good photostability, flexible luminous range, and biological compatibility. Chiu and coworkers [91] presented yellow-emission Pdots containing the BODIPY-based polymer **62** as the acceptor, poly[9,9-dioctylfluorenyl-2,7-diyl-co-1,4-benzo-(2,1′-3)-thiadiazole] (PFBT) as the donor, and poly(styrene-co-maleic anhydride) (PSMA) as the crosslinking agent (Figure 20). When the ratio of BODIPY polymers to PFBT was 0.54, the Pdots exhibited bright yellow emission based on the efficient Förster resonant energy transfer (FRET) between acceptors and donors. To realize specific cellular imaging, Pdots were further conjugated with streptavidin to obtain Pdot-SA fluorescence probe, which specifically bound biotin anti-EpCAM receptor onto the surface of MCF-7 breast cancer cells. The intensity of the Pdot-SA probe in MCF cells was about five times higher than that of the commercial Qdot 565-streptavidin bioconjugates under the same condition. Overall, Pdots based on crosslinked BODIPY-containing conjugated polymers are qualified for application as bright fluorescent biological probes.

Furthermore, Zhang and coworkers [93] reported two hyperbranched BODIPY-containing conjugated polymers **63** and **64** with 2,7-bis(4,4,5,5-tetramethyl-1,3,2-dioxaborolan-2-yl)-9,9-dioctylfluorene, 4,7-dibromobenzothiadiazole, and triiodo-BODIPY as monomers through Suzuki cross-coupling reaction (Figure 21a). Owing to the different reactive abilities of iodine and bromide with boric acid/ether, different mixing procedures were adopted. One was to add all three reactive monomers at once, and in this case, the “local” concentration of BODIPY units in the resultant hyperbranched polymer **63** was high, which would limit the fluorescence emission of **63** due to the concentration quenching effect of BODIPY units. The other was to add triiodo-BODIPYs to the reaction in batches, which resulted in self-quenching reduction. The mixtures of these polymers and PSMA were nanoprecipitated to form small and stable Pdots (**63**-Pdot and **64**-Pdot). **64**-Pdot showed a quantum yield of 22%, higher than that of the corresponding linear polymers, whereas **63**-Pdot showed a quantum yield of only 11%. The data suggested that hyperbranching was an alternative approach to restrain fluorescence quenching. Similar to the work of Chiu’s group [91], the two Pdots further modified with streptavidin could specifically label MCF-7 cells and exhibited bright fluorescence and good biocompatibility (Figure 21b). These works highlight the potential of Pdots based on different conjugated structures in bioimaging and provide an efficient method to increase brightness and decrease the quenching degree.

Recently, multifunction dyes have been highlighted for biological applications, especially the dyes that have both bioimaging and biotherapy capabilities. Zhao and coworkers [94] proposed a new ionized polyfluorene **65** conjugated with BODPIYs and NIR phosphorescent iridium(III) complexes, and polymers **66** and **67** containing fluorene and BODIPYs or iridium(III) complexes respectively, as control groups (Figure 22). Consequently, the authors could investigate the FRET process between BODIPYs donor and iridium(III) complexes acceptor. Because of the existence of ionized monomers, the conjugated polymers tended to self-assemble to establish NPs in water, and the conjugated backbone protected the NPs from photobleaching, which was favorable for enduring biological applications. Meanwhile, the BODIPYs amplified the light absorption ability of NPs, and iridium(III) complexes were responsible for effectively producing singlet oxygen based on the potent FRET. According to in vitro and in vivo experiments, the outstanding inhibition of tumor growth was induced by the high photodynamic therapy (PDT) efficiency based on the high singlet-oxygen quantum yield (97%) (Figure 22). Interestingly, taking advantage of the linear relationship between the phosphorescence lifetime of NPs or the emission intensity ratio and the O_2_ content, the authors could realize real-time O_2_ imaging and provide guidelines for PDT assessment. The rational structural design would inspire the progress of more prominent photosensitizers with complementary groups for cancer diagnosis and theranostic.

Xie and coworkers [90] developed a BODIPY-based conjugated polymer **68** containing hexadecane, whose structures were similar to those of bioactive fatty acids. The NP formulations of **68** and human serum albumin (HSA), a critical transport protein in the body, were rationally prepared through ultrasonic emulsification based on the supramolecular interaction between **68** and HSA (Figure 23). The NPs displayed enhanced water solubility, improved stability, and similar NIR absorption compared with the conjugated polymer **68**. A certain degree of aggregation quenching was conducive to more potent photothermal therapy (PTT), which inspired the investigation of the potential phototherapeutic applications of the polymer. The NPs possessed photothermal effect, as evidenced by the noticeable rise in the NP solution temperature under 685 nm laser irradiation. The experiment analysis showed that the photothermal conversion efficiency of NPs could reach 37.5%, and the photothermal reproducibility was prominent. To investigate the further application of NPs, a series of in vitro and in vivo experiments were performed. The in vitro experiment showed that the NPs were distributed in the cytoplasm by efficient cytoplasmic endocytosis and displayed significant cytotoxicity under laser radiation. In addition, the in vivo experiment results showed that PTT via intra-tumoral injection with irradiation provided the maximum tumor ablation effect, and adverse effects were almost negligible. Also, owing to their NIR absorption and high photothermal efficacy, NPs have emerged as multi-mode bioimaging materials with readable fluorescence signals and clear photoacoustic (PA) signals. Similarly, in another work [89], Xie’s group prepared a series of conjugated polymers **69**–**71** (Figure 23) from BODIPY with various numbers of methyl substituents and DPP. Subsequently, conjugated polymers and F127 were combined to obtain NPs that exhibited PTT capacity, PA imaging, and infrared thermal imaging. These studies usher in a new generation of multifunctional biomedical materials and develop image-guided therapy.

### 3.3. Sensor

Owing to the wide range of absorption and emission spectra of BODIPY derivatives and the characteristics of conjugated chain amplification signal, BODIPY-based conjugated polymers are applied to anion or organic vapor sensors [96,97,98,99]. Cao and coworkers [96] presented a new conjugated polymer **72**, synthesized through Sonogashira cross-coupling reaction between BODIPY diiodide and alkynyl fluorene, for the detection of F^−^ and CN^−^ ions (Figure 24). When F^−^ was added to the solution of **72**, the solution color changed from purple to orange by degrees, and the emission color changed from red to yellow by degrees, which was consistent with the new emission band at 579 nm. However, when CN^−^ was added to the solution of **72**, the solution color gradually changed from purple to faint yellow, and the emission color gradually changed from red to yellowish green, which agrees with the new emission bands at 514 and 563 nm. Other ions did not cause a significant change in the naked color or emission color. Thus, probe **72** could allow the fast naked-eye detection of F^−^ and CN^−^ ions without interference from other ions, and the detection limits of **72** toward F^−^ and CN^−^ ions were 5.23 × 10^−7^ and 2.96 × 10^−7^ M, respectively. Nuclear magnetic resonance (NMR) demonstrated that the excellent detection ability of **72** was derived from the nucleophilic attack of F^−^ and CN^−^ to difluoroboron bridges. Besides its use as a sensor medium, **72** was also suitable as an imaging agent with low cytotoxicity. This work not only replenishes the BODIPY-based conjugated polymer library but also provides a feasible method to sense ions through the color response caused by highly responsive structural change.

Valiyaveettil and coworkers [98] synthesized a series of hyperbranched conjugated polymers (Figure 25) based on BODIPYs for detecting organic vapors. To study the influence of polymer conformation on vapor sensing, the authors developed **73** and **74** as A_2_B_3_-type hyperbranched conjugated polymers with benzene and triphenylamine between BODIPY moieties respectively, with **75** as an A_2_B_4_-type hyperbranched conjugated polymer with pyrene between BODIPY moieties. These hyperbranched conjugated polymers all displayed solvent effect in different solvents. Polymer **73** possessed a more planar conformation, which resulted in emission quenching at solid state. The twisted conformation of **74** and the higher branching degree of **75** provided the polymers with weaker aggregation, leading to NIR emission even at the solid state. Compared with **73** and **75**, **74** exhibited higher selectivity toward aromatic solvent, and the high reproducibility. Thus, the systematic research developed a new vapor detection utilizing hyperbranched BODIPY-based conjugated polymers with better performance, enriching the structures and applications of BODIPY-based conjugated polymers.

### 3.4. Gas/Energy Storage

Generally, conjugated porous polymers (CPPs), owing to their favorable structures, have been extensively applied to gas or energy storage, and BODIPY units are rich in heteroatoms, amplifying their affinity to gas or other substances [100,101,102,103]. Patra and coworkers [103] prepared four BODIPY-based CPPs **76**–**78** (Figure 26) with various alkyl chain lengths at meso-positions via Sonogashira cross-coupling reaction. Polymer **76** possessed mesoporous structures with Brunauer-Emmett-Teller (BET) surface area of 73 (**76a**) and 322 m^2^ g^−1^ (**76b**), while **77** and **78** were respectively microporous with BET surface area of 573 m^2^ g^−1^ and ultra-microporous with BET surface area of 1010 m^2^ g^−1^. The different pore features were related to alkyl chain substitution; that is, the long alkyl chain took up more free volume, resulting in the pore blockage effect. On the contrary, rigid phenyl with nonplanarity led to small pore diameter and high surface area. Owing to profitable pore features, **78** displayed the highest CO_2_ uptake of 16.5 wt%, higher than those of **77** (10.5 wt%) and **76b** (6.9 wt%). In addition, **78** showed the highest isosteric heat of adsorption (30.6 kJ mol^−1^), due to the rich nitrogen content of BODIPYs, leading to the good affinity of CO_2_. Notably, **77** possessed the best selective adsorption to CO_2_ in the binary mixtures of CO_2_ and N_2_, because the appropriate pore size maintained stronger electrostatic interaction between POPs and CO_2_. BODIPY-based POPs have the capability of producing ^1^O_2_, which can catalyze some oxidation reactions with a quite high conversion.

Likewise, to exploit superior anode materials of lithium-ion batteries, Kuang and coworkers [102] prepared a new POP **79** (Figure 27) containing anthracene-substituted BODIPYs and 1,3,5-triethynylbenzene, and further obtained carbonized POP (**79-C**) via a carbonization process. Polymers **79** and **79-C** displayed microporous architectures with pore sizes of 20.5 and 6.9 nm respectively, and their BET surface areas were 505 and 638 m^2^ g^−1^, respectively. In addition, the existence of heteroatoms allowed POP **79** to possess higher specific capacity, better superior rate performance, and long-term cyclic stability. As for **79-C**, heteroatoms doping also made conductivity and the photoelectric property better. Hence, BODIPY-based POPs pave a way for the fabrication of gas/energy storage materials based on the porous structure and richness in heteroatoms of BODIPY units.

### 3.5. Other Applications

In addition to the above-mentioned applications, there are other fields involving BODIPY-based conjugated polymers. Yang and coworkers [104] presented a new linear conjugated polymer **80** via Suzuki cross-coupling reaction (Figure 28). Polymer **80** with the conjugated chains displayed better chemical and thermal stability compared with the monomers. To improve the dispersibility and catalytic efficiency, **80** was combined with silica powder through the F…Si interactive force, and **80-silica** was obtained (Figure 28), which was hardly soluble in common organic solvent, allowing easy recovery from reactions. A series of control experiments proved that the yield of benzylamine oxidation was up to 82% at the reaction temperature of 90 °C in the heterogeneous reaction using **80-silica** as a catalyst. Such excellent catalytic effect depended on the generation of ^1^O_2_ from **80-silica** under radiation. The recycling performance of **80-silica** was prominent. After it was used three times, the yield only decreased by 5%. Furthermore, BODIPY-based POPs with self-porosity could also serve as photocatalysts without hybridization. Liras and coworkers [105] prepared a new microporous polymer **81** with high surface area (Figure 28). The polymer could be used as a heterogeneous catalyst to oxidize thioanisole to sulfoxide with a rate four times higher than that of homogeneous catalysts. However, polymer **81** could only be reused twice. These heterogeneous photocatalysts based on BODIPY-containing polymers [104,105,106] represent a new potential class of high-catalytic-efficiency and recyclable catalysts, resulting in the simplification of handling and purification.

Given the fast reaction between the 2,6-positions of BODIPY units and iodine, Kuang and coworkers [107] synthesized a new POP **82** with 2,6-position-unsubitituted BODIPY units as monomers via Sonogashira cross-coupling reaction and verified its capability of capturing iodine compared with 2,6-positions-subitituted BODIPY-based POP **83** and the reference compound **NBDP** without BODIPYs (Figure 29). The polymerization process was accompanied by a side oxidation-reaction, which consumed sectional active sides but reduced the surface area and porous structure for a higher adsorption capacity. The comparison of **82**, **83**, and **NBDP** showed an increasing iodine absorption capacity from **NBDP** to **83** to **82**, indicating that not only porous properties but also the presence of affinity function groups impacted the iodine absorption capacity. Moreover, **82** exhibited higher adsorption capability and fast inclusion rate in iodine-hexane solution, and its release rate was also the fastest. Additionally, all POPs showed good regeneration capacity for recycling. These porous materials not only have potential capacity of iodine absorption but also provide a feasible strategy to facilitate the development of other volatile compounds.

Recently, Wu and coworkers [108] designed two BODIPY-based conjugated polymers **84** and **85** (Figure 30), and then further prepared Pdots with uniform small-size distribution for super-resolution optical fluctuation imaging (SOFI). This imaging technique is based on the random fluctuation of the fluorescence intensity of mutually fluorescent chromogens to achieve resolution improvement. Polymers **84** and **85** displayed emission maxima at 565 and 720 nm, and due to the FRET effect between polymer chains, the fluorescence spectra of these polymer materials were greatly narrow, with ~39 nm of emission bandwidths (full width at half maximum, FWHM) for **84** and ~50 nm of emission bandwidths (FWHM) for **85**. Compared with commercially available fluorescent dyes Alexa 555 and Alexa 647, **84**-Pdot and **85**-Pdot exhibited stronger luminescence, higher signal-to-noise ratio, excellent single-particle anti-photobleaching properties, and more pronounced fluctuations; thus, they can be applied for SOFI. Indeed, the single-particle analysis demonstrated that after eight-order SOFI processing, the resolution of the Pdots SOFI imaging significantly amplified to 57 (**84**) and 88 nm (**85**). Furthermore, after surface bio-conjugated with streptavidin, the Pdots achieved the specific labeling of the mitochondrial membrane structure and microtubule structure of BS-C-1 without cross-color interference and the eighth-order SOFI imaging. Consequently, the SOFI imaging system based on BODIPY-containing conjugated polymers can promote the development of higher-resolution imaging systems in the future.

## 4. Conclusions

Significant advances have been achieved in BODIPY-based conjugated polymers due to the joint effect of conjugate structures and the fluorescent dye BODIPY, which encouraged us to comprehensively summarize the establishments and applications of BODIPY-based conjugated polymers. BODIPYs possess the advantages of facile synthesis and chemical structure modification and can be synthesized through the conventional preparation methods of conjugated polymers; among the preparation methods, metal-catalyzed cross-coupling polymerization is most frequently used, especially Suzuki cross-coupling or Sonogashira cross-coupling reaction. Due to the numerous active sites of the BODIPY core, the structures of BODIPY-based conjugated polymers are flexible, comprising linear, coiled, porous, branched, and crosslinked structures; among these, linear and coiled conjugated polymers polymerized through α-α-connected, α-β-connected, or β-β-connected account for the largest proportion. In this respect, we summarized design of new polymers that can be initiated from three aspects: (1) the rational selection of polymerization sites, (2) the modification of BODIPY building units before polymerization, and (3) the appropriate selection of comonomers. Furthermore, some significant conclusions are as follows. First, π-conjugation is more efficient in the case of polymerization through the α-position of BODIPYs than in the polymerization through the β-position, allowing for low band gap. Second, the extension of conjugated BODIPYs at the 3,5-position through classical Knoevenagel condensation reaction results in deep-red or NIR emission. Third, D-A structures promote intramolecular charge transfer, leading to low band gap and broad absorption. Fourth, the incorporation of BODIPYs as side chains into polymers can effectively inhibit the concentration fluorescence quenching and even result in FERT. Finally, BODIPY-based conjugated polymers possess high flexibility in the design of skeletons and porosity. Owing to flexible design and diverse structures, conjugated polymers containing BODIPYs exhibit remarkable potential for applications in several fields, including optoelectronic materials, biotherapy, bioimaging, sensors, gas/energy storage, and photocatalysis. They are most widely utilized in photoelectric materials, especially solar cells, owing to the excellent photoelectric properties both of BODIPY dyes and conjugated backbones.

Furthermore, significant achievements have been made in the development of BODIPY-based conjugated polymers, from structural design to practical applications. However, there is still much room for further growth. Moreover, water solubility and NIR emission are imperative for biomedical applications. Despite that some water-soluble polymeric BODIPYs modified by the introduction of hydrophilic chains or biological molecules have been obtained, there are limited cases and no certain regulations for designing suitable polymers; thus, this area is an open proposition. For greater tissue penetration and less biological damage, NIR region II absorption and emission are the better choice, which is a challenging direction. In addition, designing new BODIPY-based conjugated POPs with higher BET surface areas, anti-degradation backbones, and specific binding groups may afford the BODIPY the capability to adsorb or store more chemical substances and energy. Furthermore, POPs containing BODIPYs are promising electron donor scaffolds for electron-accepting compounds, and their exceptional light-absorption properties and high-dimensional geometry are conducive to amplified charge generation, splitting, and transport. These advantages indicate that POPs are reasonable candidates for photoelectric materials, but there are few related examples and linear polymers are in the majority. It may be that solubility restricts the POP applications; therefore, certain effort should be channeled toward designing rational POPs and uniform film formation to integrate them into photoelectric devices. Moreover, BODIPY-containing conjugated polymers have been applied in photocatalysis, due to the generation of ^1^O_2_ with the action of BODIPYs. It is a question of whether these polymers with energy class levels are allowed to catalyze reduction reactions, such as hydrogen reduction, when they agree with the oxidation and reduction potentials required for the reaction. It can be anticipated that BODIPY-based conjugated polymers may continually broaden the application range of photoelectric materials, biological materials, and other smart materials. This review can inspire chemists, material scientists, and engineers to jointly develop new BODIPY-based conjugated polymers and explore their broader practical applications.

## Figures and Tables

**Figure 1 polymers-13-00075-f001:**
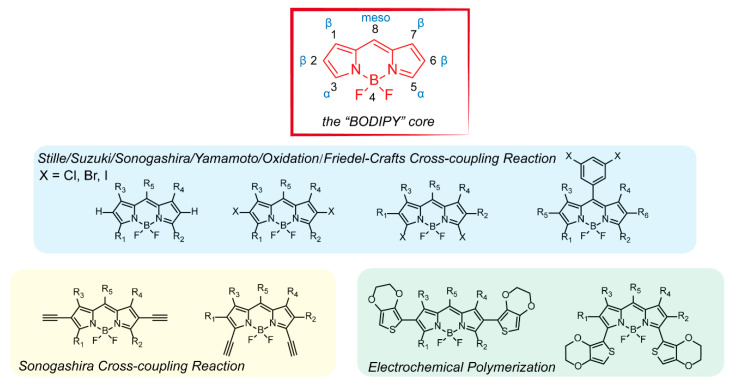
Structural representation of the 4,4′-Difluoro-4-bora-3a,4a-diaza-s-indacene (BODIPY) core and BODIPY intermediates suited to polymerization via transition metal–catalyzed polycondensation reactions or electrochemical polymerization.

**Figure 2 polymers-13-00075-f002:**
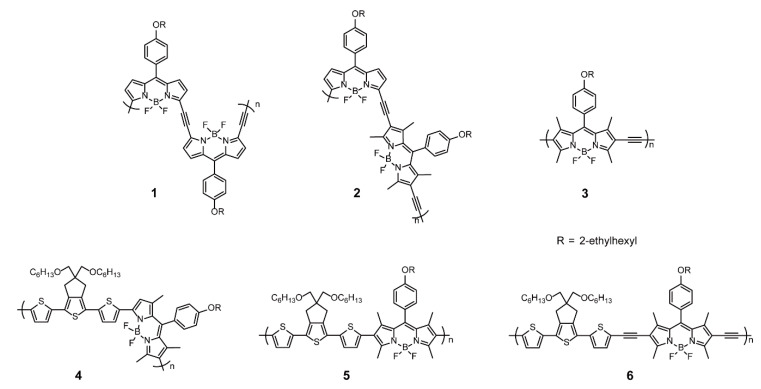
Chemical structures of polymers **1–6** (Reproduced with permission from Reference [30]. Copyright @ 2016, John Wiley and Sons. Reproduced with permission from Reference [31]. Copyright @ 2015, American Chemical Society).

**Figure 3 polymers-13-00075-f003:**
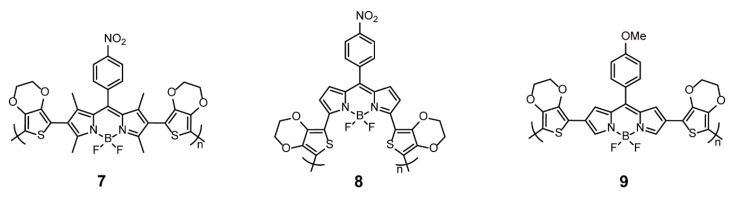
Chemical structures of polymers **7–9** (Reproduced with permission from Reference [23]. Copyright @ 2009, Elsevier. Reproduced with permission from Reference [25]. Copyright @ 2009, American Chemical Society. Reproduced with permission from Reference [33]. Copyright @ 2014, Elsevier).

**Figure 4 polymers-13-00075-f004:**
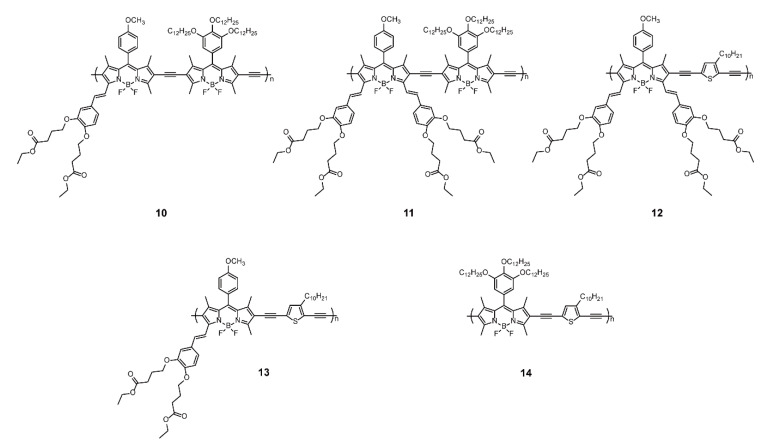
Chemical structures of polymers **10–14** (Reproduced with permission from Reference [34]. Copyright @ 2012, Royal Society of Chemistry).

**Figure 5 polymers-13-00075-f005:**
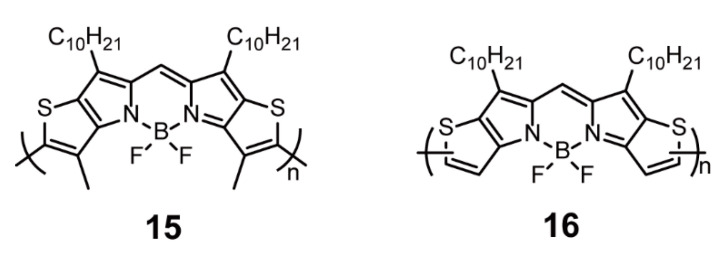
Chemical structures of polymers **15** and **16** (Reproduced with permission from Reference [40]. Copyright @ 2014, American Chemical Society).

**Figure 6 polymers-13-00075-f006:**
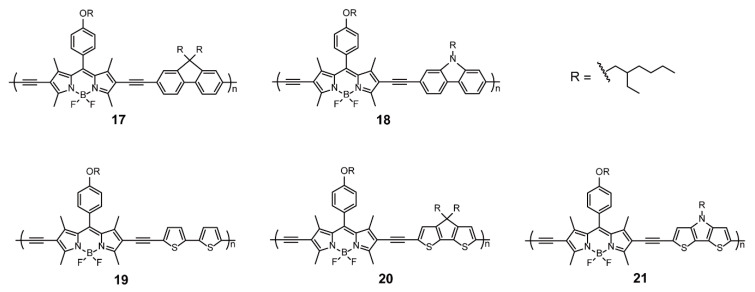
Chemical structures of polymers **17–21** (Reproduced with permission from Reference [48]. Copyright @ 2012, Royal Society of Chemistry).

**Figure 7 polymers-13-00075-f007:**
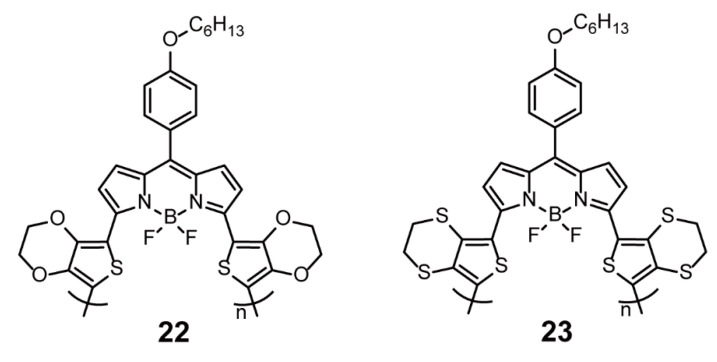
Chemical structures of polymers **22** and **23** (Reproduced with permission from Reference [49]. Copyright @ 2012, Royal Society of Chemistry).

**Figure 8 polymers-13-00075-f008:**
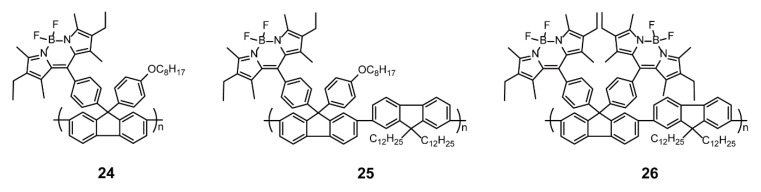
Chemical structures of polymers **24**–**26** (Reproduced with permission from Reference [54]. Copyright @ 2013, American Chemical Society).

**Figure 9 polymers-13-00075-f009:**
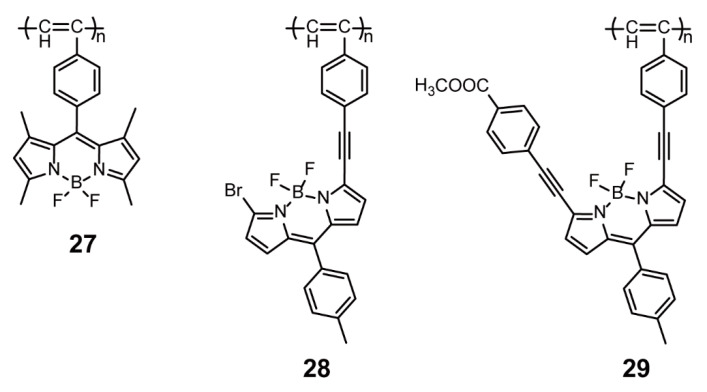
Chemical structures of polymers **27**–**29** (Reproduced with permission from Reference [55]. Copyright @ 2014, Royal Society of Chemistry).

**Figure 10 polymers-13-00075-f010:**
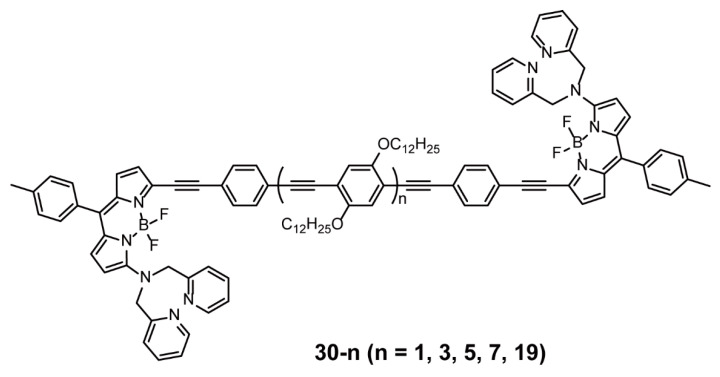
Chemical structure of polymer **30** (Reproduced with permission from Reference [56]. Copyright @ 2011, Elsevier. Reproduced with permission from Reference [57]. Copyright @ 2011 WILEY-VCH Verlag GmbH & Co. KGaA, Weinheim.).

**Figure 11 polymers-13-00075-f011:**
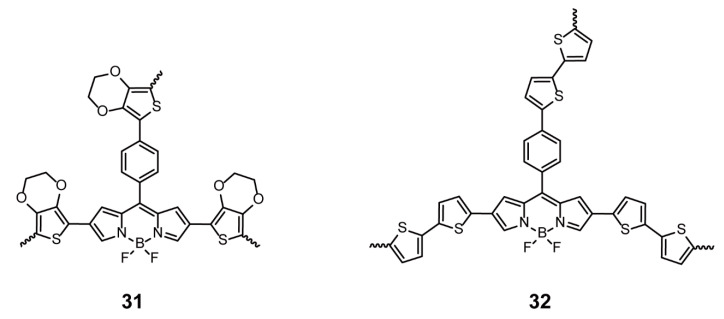
Chemical structures of polymers **31** and **32** (Reproduced with permission from Reference [24]. Copyright @ 2013 Elsevier Ltd.).

**Figure 12 polymers-13-00075-f012:**
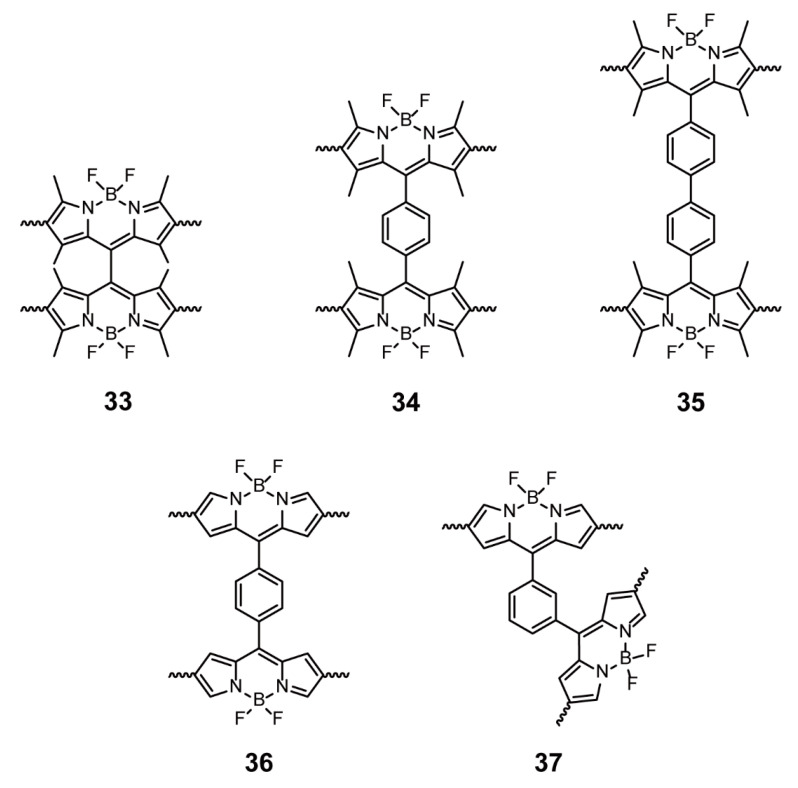
Chemical structures of polymers **33**–**37** (Reproduced with permission from Reference [65]. Copyright @ 2017 WILEY-VCH Verlag GmbH & Co. KGaA, Weinheim. Reproduced with permission from Reference [66]. Copyright @ 2017, Royal Society of Chemistry).

**Figure 13 polymers-13-00075-f013:**
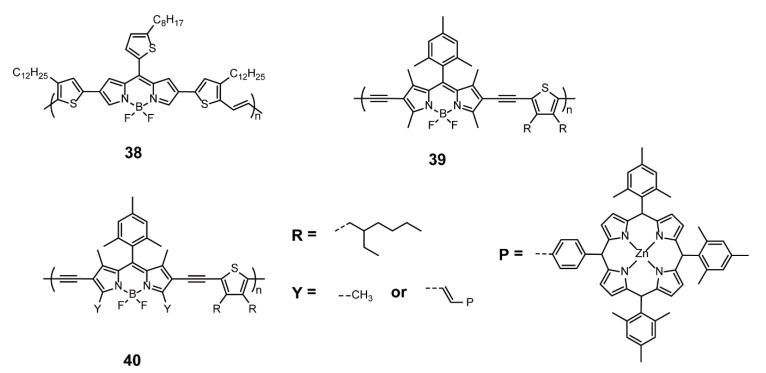
Chemical structures of polymers **38**–**40** (Reproduced with permission from Reference [74]. Copyright @ 2018, American Chemical Society. Reproduced with permission from Reference [75]. Copyright @ 2015, Royal Society of Chemistry).

**Figure 14 polymers-13-00075-f014:**
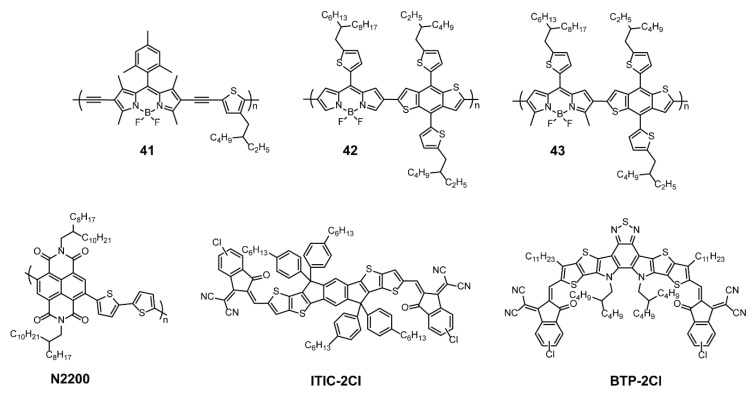
Chemical structures of polymers **41**–**43** and non-fullerenes N2200, ITIC-2Cl, and BTP-2Cl (Reproduced with permission from Reference [76]. Copyright @ 2018, American Chemical Society. Reproduced with permission from Reference [77]. Copyright @ 2019, American Chemical Society. Reproduced with permission from Reference [78]. Copyright @ 2019, Royal Society of Chemistry).

**Figure 15 polymers-13-00075-f015:**
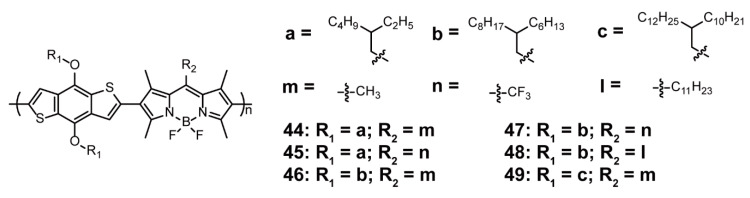
Chemical structures of polymers **44**–**49** (Reproduced with permission from Reference [81]. Copyright @ 2018, American Chemical Society).

**Figure 16 polymers-13-00075-f016:**
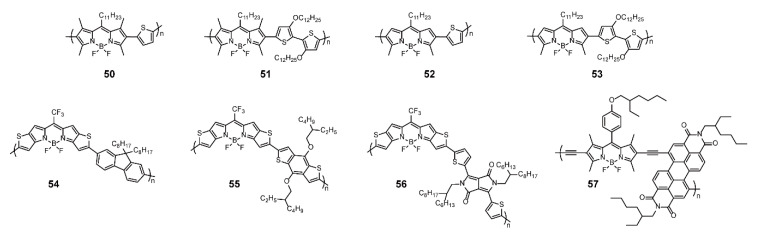
Chemical structures of polymers **50**–**57** (Reproduced with permission from Reference [82]. Copyright @2013 WILEY-VCH Verlag GmbH & Co. KGaA, Weinheim. Reproduced with permission from Reference [83]. Copyright @ 2020, American Chemical Society. Reproduced with permission from Reference [84]. Copyright @ 2011, American Chemical Society).

**Figure 17 polymers-13-00075-f017:**
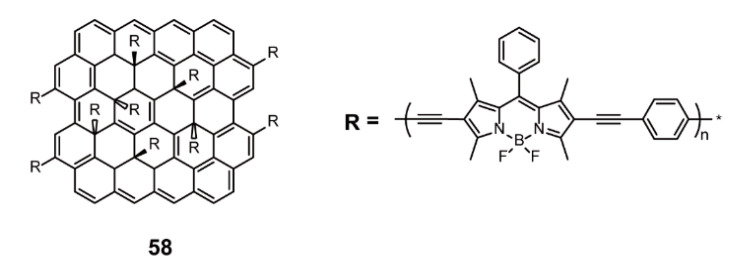
Chemical structures of polymer **58** (Reproduced with permission from Reference [87]. Copyright @ 2017 Elsevier Ltd).

**Figure 18 polymers-13-00075-f018:**
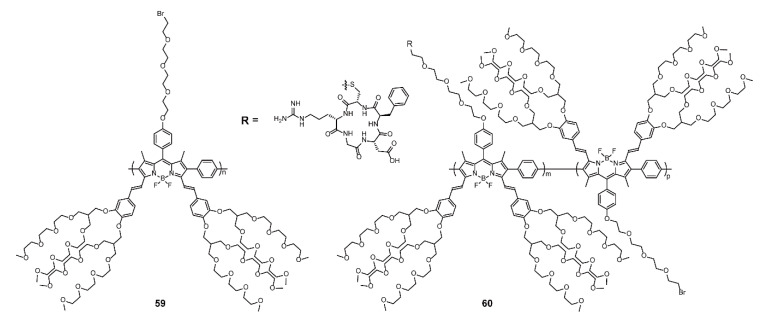
Chemical structures of polymers **59** and **60** (Reproduced with permission from Reference [92]. Copyright @ 2012 Elsevier B.V.).

**Figure 19 polymers-13-00075-f019:**
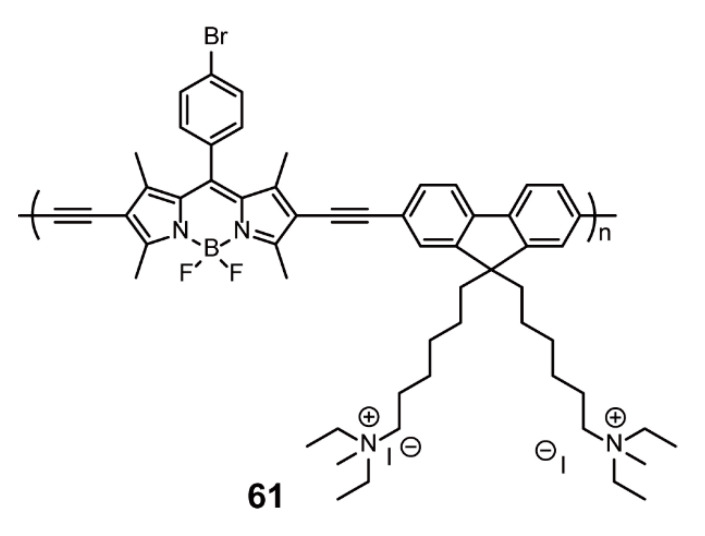
Chemical structure of polymer **61** (Reproduced with permission from Reference [95]. Copyright @ 2015, Springer Science Business Media New York).

**Figure 20 polymers-13-00075-f020:**
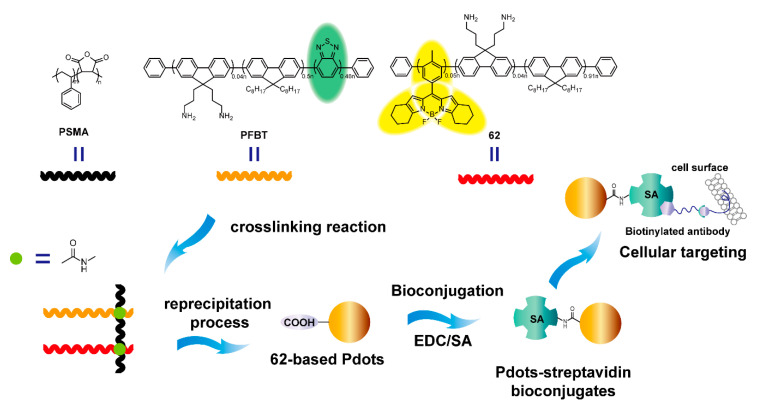
Schematic illustration of the preparation of crosslinked semiconducting polymers with poly(styrene-co-maleic anhydride) (PSMA), poly[9,9-dioctylfluorenyl-2,7-diyl-co-1,4-benzo-(2,1′-3)-thiadiazole] (PFBT), and **62**, and **62**-based polymer dots (Pdot) bioconjugates for specific cellular targeting (Reproduced with permission from Reference [91]. Copyright @ 2014 ACS publication).

**Figure 21 polymers-13-00075-f021:**
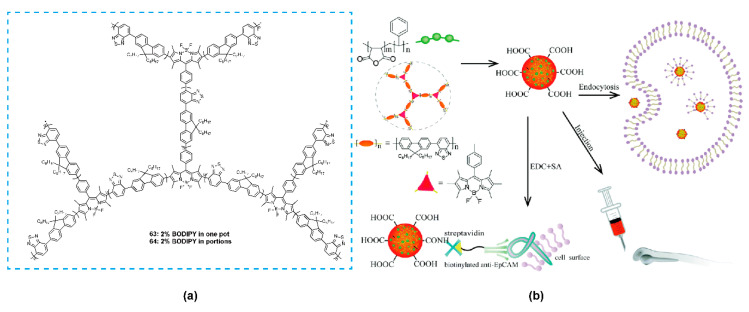
(**a**) Chemical structures of polymers **63** and **64**. (**b**) Illustration of the hyperbranched BODIPY-based Pdots and their bioconjugate probes for cellular imaging (Reproduced with permission from Reference [93]. Copyright @ 2017 Royal Society of Chemistry).

**Figure 22 polymers-13-00075-f022:**
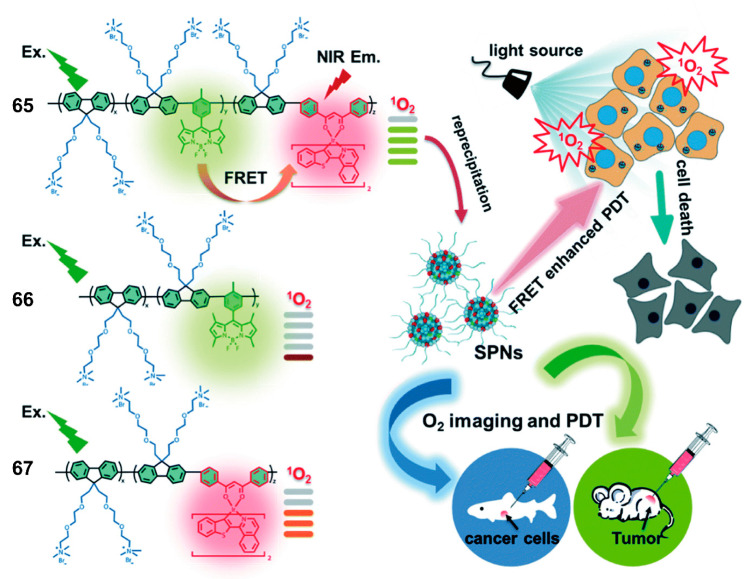
Chemical structures of polymers **65**–**67** and graphical representation of **65** used as photosensitizers with enhanced ^1^O_2_ generation through Förster resonant energy transfer (FRET) for efficient tumor treatment (Reproduced with permission from Reference [94]. Copyright @ 2019 Royal Society of Chemistry).

**Figure 23 polymers-13-00075-f023:**
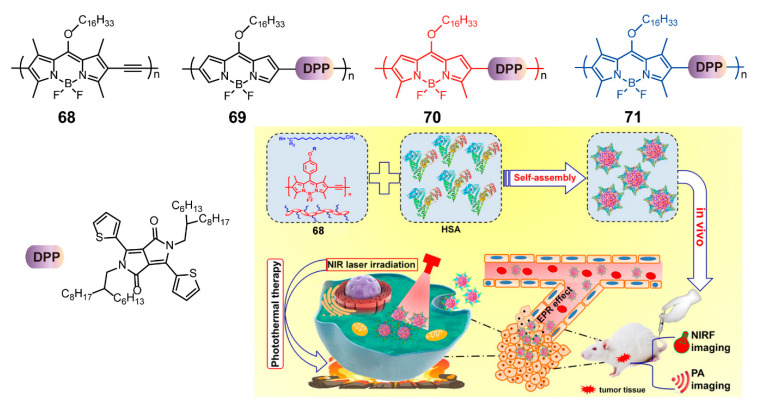
Chemical structures of polymers **68–71** and schematic illustration of fabrication self-assembly procedures and applications of NPs (Reproduced with permission from Reference [90]. Copyright @ 2019 ACS publication).

**Figure 24 polymers-13-00075-f024:**
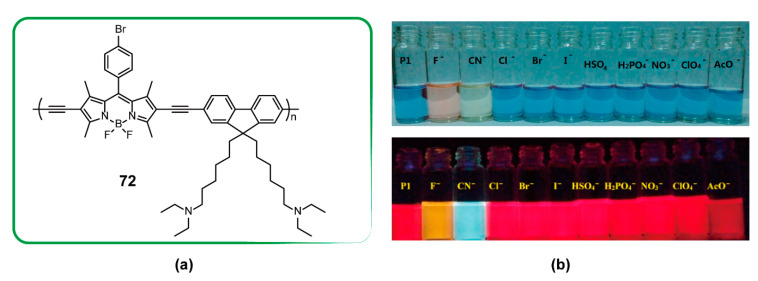
(**a**) Chemical structures of polymer **72**. (**b**) Visual color changes and emission changes (λ_ex_ = 365 nm) of **72** (c = 10 μM) in tetrahydrofuran/H_2_O (98:2, *v*/*v*) in the presence of different anions (Reproduced with permission from Reference [96]. Copyright @ 2015 Elsevier).

**Figure 25 polymers-13-00075-f025:**
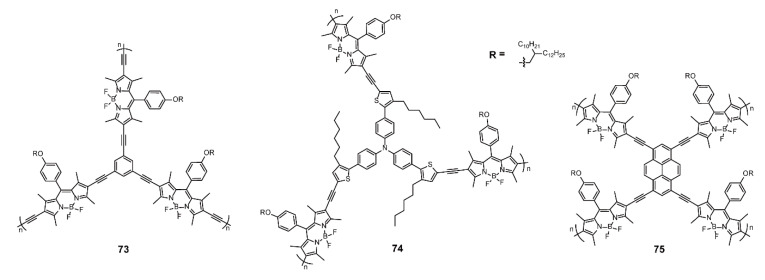
Chemical structures of polymers **73–75 [98]** (Reproduced with permission from Reference [96]. Copyright @ 2016, Royal Society of Chemistry).

**Figure 26 polymers-13-00075-f026:**
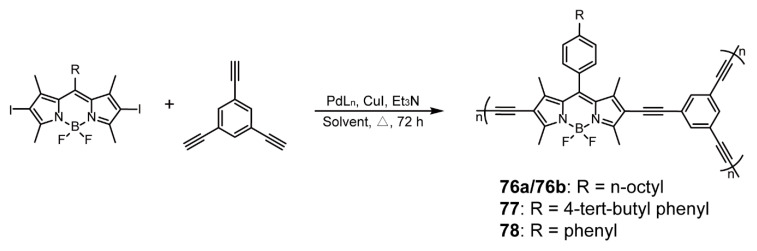
Synthesis of polymers **76–78**. (Reaction conditions: **76a:** [Pd(PPh_3_)_2_Cl_2_], toluene, 80 °C; **76b**: [Pd(PPh_3_)_4_], dimethylformamide, 130 °C; **77**: M2 monomer, [Pd(PPh_3_)_4_], DMF, 130 °C; **78**: M3 monomer, [Pd(PPh_3_)_4_], DMF, 130 °C) (Reproduced with permission from Reference [103]. Copyright @ 2016, American Chemical Society).

**Figure 27 polymers-13-00075-f027:**
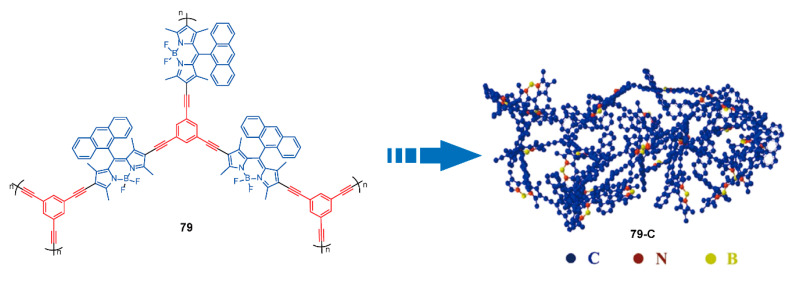
Chemical structures of polymers **79** and diagram of **79-C** (Reproduced with permission from Reference [102]. Copyright @ 2018 ACS publication).

**Figure 28 polymers-13-00075-f028:**
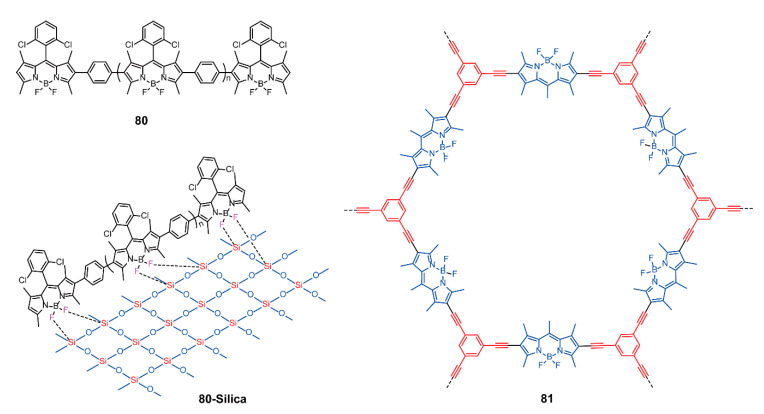
Chemical structures of polymers **80**, **80-silica**, and **81** (Reproduced with permission from Reference [104]. Copyright @ 2015 Elsevier B.V. Reproduced with permission from Reference [105]. Copyright @ 2016, American Chemical Society).

**Figure 29 polymers-13-00075-f029:**
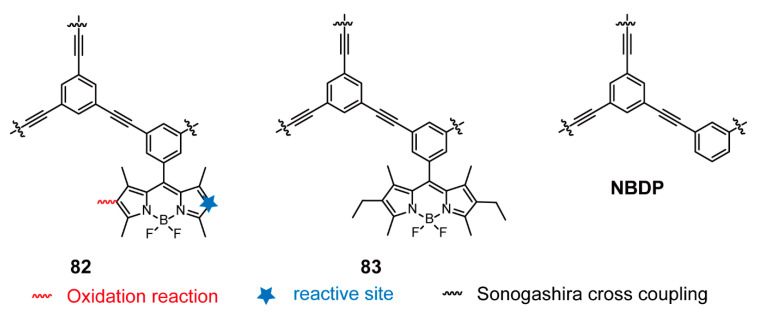
Chemical structures of polymers **82**, **83**, and **NBDP** (Reproduced with permission from Reference [107]. Copyright @ 2017, Royal Society of Chemistry).

**Figure 30 polymers-13-00075-f030:**
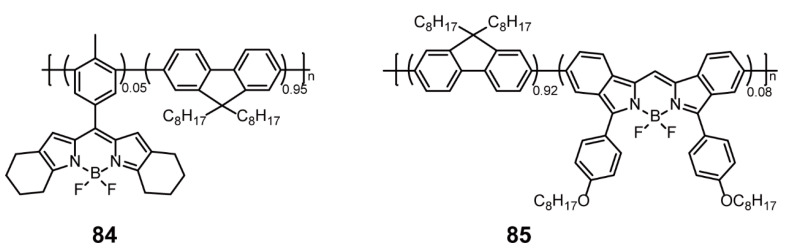
Chemical structures of polymers **84** and **85** (Reproduced with permission from Reference [108]. Copyright @ 2016, Royal Society of Chemistry).

**Table 1 polymers-13-00075-t001:** Summary of the structures and synthetic methods of BODIPY-based conjugated polymers.

Structures of BODIPY-Based Conjugated Polymers		Synthetic Methods	Compounds	References
Linear/Coiled Conjugated Polymers	BODIPY as Backbone Units	Stille Cross-coupling Reaction	**4**, **5**, **22**, **23**	[31,49]
		Sonogashira Cross-coupling Reaction	**1**–**3**, **6**, **10**–**14**, **17**–**21**	[30,31,34]
		Oxidation Coupling	**15**, **16**	[40]
		Electrochemical Polymerization	**7**–**9**	[23,25,33]
	BODIPY as Pendants Side Chains	Suzuki Cross-coupling Reaction	**24**–**26**	[54]
		Coordination Polymerization	**27**–**29**	[55]
	BODIPY as End Groups	Sonogashira Cross-coupling Reaction	**30-n**	[56,57]
Porous Conjugated Polymers		Electrochemical Polymerization	**31**, **32**	[24]
		Friedel-Crafts cross-coupling reactions	**33**–**37**	[65,66]
Other Structures	Crosslinked Polymers	Yamamoto Cross-coupling Reaction	**62**	[91]
	Branched Polymers	Suzuki Cross-coupling Reaction	**63**, **64**	[93]
		Sonogashira Cross-coupling Reaction	**73**–**75**	[98]

**Table 2 polymers-13-00075-t002:** Summary of the applications and synthetic methods of BODIPY-based conjugated polymers.

Functional Applications of BODIPY-Based Conjugated Polymers		Synthetic Methods	Compounds	References
Optoelectronic Materials	Solar Cells	Stille Cross-coupling Reaction	**38**, **42**–**49**	[75,77,78,81]
		Sonogashira Cross-coupling Reaction	**39**–**41**	[74,76]
	Organic Thin-film Transistors	Stille Cross-coupling Reaction	**50**–**56**	[82,83]
		Sonogashira Cross-coupling Reaction	**57**	[84]
	Memory Devices	Sonogashira Cross-coupling Reaction	**58**	[87]
Bioimaging and Biotherapy		Suzuki Cross-coupling Reaction	**59**, **60**, **63**–**66**, **69**–**91**	[90,92,93,94]
		Sonogashira Cross-coupling Reaction	**61**, **68**	[90,95]
		Yamamoto Cross-coupling Reaction	**62**	[91]
Sensor		Sonogashira Cross-coupling Reaction	**72**–**75**	[96,98]
Gas/Energy Storage		Sonogashira Cross-coupling Reaction	**76**–**79**	[102,103]
Other Applications		Suzuki Cross-coupling Reaction	**80**	[104]
		Sonogashira Cross-coupling Reaction	**81**–**83**	[105,107]
		Yamamoto Cross-coupling Reaction	**84**, **85**	[108]

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
