# Peer review of "Architectures and Applications of BODIPY-Based Conjugated Polymers"

_polymers, 2020, doi:10.3390/polym13010075_

Round 1

Reviewer 1 Report

1: Please modify the title as : Architectures and Applications of BODIYP-based Cnojugated Polymers: A review

2: IN the abstract the authors state that  In2008,174,4-difluoro-4-bora-3a,4a-diazs-indacene(BODIPY)was first incorporated into conjugated polymers owing to its conjugated structure and intriguing opticalproperties,-------- what does it mean? the monomer was polymerized or it was incorporated as a moitee in a conjugated polymer? this sentence is confusing and somehow misleading. Please clarify or remove.

3: In the introduction once again the autohrs state that To develop conjugated materials with better   performance,a large number of aromatic units have been successfully introduced to conjugated polymers, including phenylacetylene, thiophene, carbazole, fluorene,and 434,4-difluoro-4-bora-3a,4a-diaza-s-indacene(BODIPY). I am not able to understand what is the need of introduction of aromatic units to these conjugated polymers while these mentioned polymers are themselves aromatic and conjugated??

4: A lot of structures have been presented in Fig. 1-19 , 26-30, without reffering to the paper where originally published. Did te authors take permission for reproducing these images. Reference sholud be provided with every figure if it is adoted from a publication.

5: For a review, it is impressive if the data is summarized in tabulated form where possible. For example the applications of said materials in different fields alon with mode of application and outcomes.

Author Response

Point 1: Please modify the title as: Architectures and Applications of BODIYP-based Conjugated Polymers: A review

Response: Thank you for your suggestion. The title has been modified to “Architectures and Applications of BODIYP-based Conjugated Polymers: A Review”. Please see Line 2, Page 1.

Point 2: In the abstract the authors state that In 2008, 4,4’-difluoro-4-bora-3a,4a-diazs-indacene (BODIPY) was first incorporated into conjugated polymers owing to its conjugated structure and intriguing optical properties, what does it mean? the monomer was polymerized or it was incorporated as a moitee in a conjugated polymer? this sentence is confusing and somehow misleading. Please clarify or remove.

Response: Thank you for your suggestion. We have modified this sentence to “4,4’-Difluoro-4-bora-3a,4a-diaza-s-indacene (BODIPY)-based conjugated polymers have been also prepared owing to its conjugated structure and intriguing optical properties, including high absorption coefficients, excellent thermal/photochemical stability, and high quantum yield.” Please see Line 18, Page 1.

Point 3: In the introduction once again the authors state that To develop conjugated materials with better performance, a large number of aromatic units have been successfully introduced to conjugated polymers, including phenylacetylene, thiophene, carbazole, fluorene, and 4,4’-difluoro-4-bora-3a,4a-diaza-s-indacene (BODIPY). I am not able to understand what is the need of introduction of aromatic units to these conjugated polymers while these mentioned polymers are themselves aromatic and conjugated??

Response: Thank you for your suggestion. The sentence has been corrected into “To develop conjugated materials with better performance, a large number of functional units have been successfully introduced to polymer systems, including phenylacetylene [10], thiophene [11], carbazole [12], fluorene [13], and 4,4’-difluoro-4-bora-3a,4a-diaza-s-indacene (BODIPY) [14]. ” Please see Line 41, Page 1.

Point 4: A lot of structures have been presented in Fig. 1-19, 26-30, without referring to the paper where originally published. Did the authors take permission for reproducing these images? Reference should be provided with every figure if it is adopted from a publication.

Response: The structures presented in Fig. 1 have been designed by ourselves and the structures presented in Fig. 2-19, 26-30 have been drawn by ourselves. Thus, we think these Figs. don’t need the reproducing permission. Of course, to omit the problem of the reproducing permission, we added the references on the every Figure. Please see all the Figure.

Point 5: For a review, it is impressive if the data is summarized in tabulated form where possible. For example, the applications of said materials in different fields alon with mode of application and outcomes.

Response: Thank you for your suggestion. We have added two tables to summarize the synthesis methods, applications, and references. Please see Table 1 and 2 in Line 2, Page 4, and Line 2, Page 12.

Reviewer 2 Report

Dear Editor, dear Authors, Yiqi Fan et al. submitted review paper on the synthesis, architecture and applications of 4,4’-difluoro-4-bora-3a,4a-diaza-s-indacene (BODIPY) based fluorophores conjugated polymers. The authors presented the different strategies for the functionalization of BODIPY and their insertion in conjugated polymers. They focused mainly on linear, coiled (BIDIPY as pendent group side chain, terminal group), and porous structures of BODIPY- based conjugated polymers, as well as their structure–property relationship. Moreover, they have discussed their applications in optoelectronic devices, sensors, gas/energy storage, bioimaging and therapy. For my opinion, the manuscript content is of high interest, and well written. Moreover, it include 108 references with the examples used are well explained and represented with detailed Figures. Therefore, I recommend acceptance of the manuscript for publication in Polymers Journal. I have a suggestion to the authors below.

  • Page 3 Figure 1. For my opinion the authors should write few lines on each method represented here in the text, with a reference on each. Otherwise, they can maybe include a table summarizing the synthesis methods, applications and references.

Sincerely Yours,

Author Response

Point 1: Page 3 Figure 1. For my opinion the authors should write few lines on each method represented here in the text, with a reference on each. Otherwise, they can maybe include a table summarizing the synthesis methods, applications, and references.

Response: Thank you for your suggestion. We have provided supplementary instruction about the synthetic method. Please see Line 24, Page 2. Furthermore, we have added two tables to summarize the synthesis methods, applications, and references. Please see Table 1 and 2 in Line 2, Page 4, and Line 2, Page 12.